# Drinking Refined Deep-Sea Water Improves the Gut Ecosystem with Beneficial Effects on Intestinal Health in Humans: A Randomized Double-Blind Controlled Trial

**DOI:** 10.3390/nu12092646

**Published:** 2020-08-31

**Authors:** Hiroaki Takeuchi, Keiro Higuchi, Yu Yoshikane, Ryo Takagi, Shinji Tokuhiro, Koichi Takenaka, Wataru Oboshi, Asako Kimura, Jahirul Md. Islam, Ayami Kaneko, Shouichi Sato, Satoshi Ishizuka

**Affiliations:** 1Department of Medical Laboratory Sciences, Health and Sciences, International University of Health and Welfare Graduate School, 4-3 Kouzunomori, Narita-City 286-8686, Chiba, Japan; oboshi@iuhw.ac.jp (W.O.); a-kimura@iuhw.ac.jp (A.K.); jahir3683@gmail.com (J.M.I.); 1657021@g.iuhw.ac.jp (A.K.); s-shouichi@iuhw.ac.jp (S.S.); 2Department of Kochi Medical School, Kochi University, Kohasu, Oko-cho, Nankoku-City 783-8505, Kochi, Japan; rtakagi@ako-kasei.co.jp (R.T.); jm-tokuhirosh@kochi-u.ac.jp (S.T.); 3Center for Regional Collaboration, Kochi University, 2-17-47 Asakurahonmachi, Kochi-City 780-8073, Kochi, Japan; khiguchi@river.ocn.ne.jp (K.H.); zuka@kochi-u.ac.jp (S.I.); 4Department of Human Living Sciences, Notre Dame Seishin University, 2-16-9 Ifuku-cho, Kita-ku, Okayama-City 700-8516, Okayama, Japan; yyoshikane@m.ndsu.ac.jp; 5DyDo-T Beverage Co. Ltd., 1310-1 Hanechou-ko, Muroto-City 781-6741, Kochi, Japan; takenaka@dt-beverage.com

**Keywords:** healthy life, refined deep-sea water, abdominal discomfort, short-chain-fatty-acid, isoflavone, gut ecosystem

## Abstract

World health trends are focusing on a balanced food and beverage intake for healthy life. Refined deep-sea water (RDSW), obtained from deep-sea water collected offshore in Muroto (Japan), is mineral-rich drinking water. We previously reported that drinking RDSW improves human gut health. Here, we analyzed the effect of drinking RDSW on the gut ecosystem to understand this effect. This was a randomized double-blind controlled trial. Ninety-eight healthy adults were divided into two groups: RDSW or mineral water (control). The participants consumed 1 L of either water type daily for 12 weeks. A self-administered questionnaire and stool and urine samples were collected through the intervention. The following were determined: fecal biomarkers of secretory immunoglobulin A (sIgA), five putrefactive products, and nine short-chain-fatty-acids (SCFAs) as the primary outcomes; and three urinary isoflavones and the questionnaire as secondary outcomes. In post-intervention in the RDSW group, we found increased concentrations of five SCFAs and decreased concentrations of phenol and sIgA (*p* < 0.05). The multiple logistic analysis demonstrated that RDSW significantly affected two biomarkers (acetic and 3-methylbutanoic acids) of the five SCFAs mentioned above (*p* < 0.05). Similarly, the concentrations of urinary isoflavones tended to increase in post-intervention in the RDSW group. Constipation was significantly alleviated in the RDSW group (94%) compared with the control group (60%). Drinking RDSW improves the intestinal environment, increasing fecal SCFAs and urinary isoflavones, which leads to broad beneficial effects in human.

## 1. Introduction

Current world trends in healthy life in human focus on a balanced intake of safe foods and beverages. We purchase many foods and beverages on the market. However, some such products, promoted without suitable verification, may not exert any beneficial effect or may even be harmful.

Deep-sea water (DSW) is obtained from depths of over 200 m. It is highly pure, with low temperature stability, high mineral concentration, and active nutritional species [1]. Bottled commercial DSW-based drinking water obtained by different methods, e.g., desalinization, is available on the market. According to studies involving animal models, the DSW-based drinking water exerts hypocholesterolemic and antiatherogenic effects [2], probably by modulating lipid metabolism [3,4]. It exerts additional beneficial effects including inhibition of hyperlipemia [2], hypertension [5], obesity [6], diabetic mellitus [7], duodenal ulcer [8], and cataract formation [9,10].

We previously performed clinical trials to investigate the effects of refined deep-sea water (RDSW), a commercial bottled DSW-based drinking water, made from DSW collected offshore in Muroto (Kochi Prefecture, Japan) in humans. RDSW is widely consumed in Japan for reasons beyond simple hydration. Clinical trials confirmed diverse beneficial effects of RDSW in humans in broad areas such as hemorheology, allergy, immunology, and infectious diseases (e.g., anti-*Helicobacter pylori* activity) [11,12,13,14,15,16,17]. No adverse effects of even long-term consumption (one year) of 1 L/day were reported in any clinical trials. RDSW alleviates constipation and abdominal discomfort, and is also safe for concurrent administration with pharmaceutical agents [18]. Hence, drinking RDSW is beneficial for humans and its use is expanding to include the cosmetic industry and public health. However, the mechanism through which RDSW exerts such various beneficial effects in human is unclear.

The gut ecosystem is composed of microbiota, microbiota-derived metabolites, and the ingesta, and incorporates the microbe–microbe and host–microbe interactions. It plays a fundamental role in the well-being of human and influences the development of various diseases, including neuropsychiatric disorders [19,20,21,22,23,24]. Many factors of human life, such as the working and living conditions, and diet, affect the gut ecosystem. Probiotics, prebiotics, and functional and/or healthy foods, including fabaceous meals, are widely consumed to maintain and support the intestinal environment. Similar, there is a growing research interest worldwide into fecal microbiota transplantation, as a natural clinical treatment for gastrointestinal diseases that modulates the gut microbiota [25,26,27]. The application of fecal microbiota transplantation has been expanded to various clinical fields, such as autoimmune disease, hepatitis, metabolic syndrome, and mental disorders [28,29,30,31].

Gut microbes produce metabolites, such as isoflavones. Equol is a metabolite of the soy isoflavone daidzein. It exerts the highest physiological effect of all isoflavones on the host health. The production of equol is limited by the existence of specific bacteria and therefore, equol levels depend on the gut microenvironment [32]. One of the current research topics is how the gut environment could be supported to enhance equol production.

Collectively, the above indicate that the gut ecosystem plays an important role in human, like any organs in the body, critically influencing the human health. The aim of the current clinical trial was to investigate whether RDSW influences the biomarkers associated with the human gut ecosystem, to provide new insights into the understanding of the broad beneficial effects of drinking RDSW.

## 2. Materials and Methods

### 2.1. Clinical Study Design

The study was designed as a randomized double-blind controlled trial to compare the human gut ecosystem in individuals drinking RDSW or mineral water using a self-administered questionnaire and stool and urine analyses. The study was performed in 2016–2018 at Muroto, Kochi, Japan. The study, having severe restrictions on term and budget, was approved as feasible by the ethics committee of Kochi University (approval number 28-93), and was therefore conducted in accordance with the ethical standards laid down in the 1964 Declaration of Helsinki and its later amendments. The questionnaire and stool and urine samples were collected through the experiment (pre- and post-intervention). The clinical trial was registered in Japan Primary Registries Network (JPRN) (UMIN000022002 2016/04/20). We focused on the changes in fecal biomarker concentrations through the intervention in this study as the primary outcome, and the questionnaire (the improvement of constipation) and urine biomarker concentrations as the secondary outcomes.

### 2.2. Subjects

A total of 115 participants were enrolled in this clinical study (industry–academia–government collaboration project) in Muroto, Kochi, Japan (Figure 1). We gathered the healthy adults; we excluded participants with any illnesses and any prescription and commercial drugs use. The pregnant persons and supplement users were also excluded. After obtaining written informed consent, 98 healthy adults were randomly divided into two groups: a group drinking RDSW (*n* = 49; mean age: 48 years; range: 27–72 years; 19 men and 30 women) and a group drinking mineral water (*n* = 49; mean age: 44 years; range 22–71 years; 21 men and 28 women). The latter group was the control group. The characteristics of participants in this clinical study are shown in Table 1. The differences in pre-intervention items such as age, sex, body mass index (BMI), and biomarker concentrations between the two groups allocated were analyzed using the Mann–Whitney U test. These showed no statistical differences and these data were used in the following study.

### 2.3. Ingestion Schedule

Bottled RDSW, Amami no Mizu, Hardness 1000 (Ako Kasei Co., Ltd., Hyogo, Japan) [15], and mineral water, the most popular water consumed in Japan, were used. Both waters include no calory, protein, fat, carbohydrate and vitamin. RDSW and mineral water were hard- and soft-type water, respectively. We purchased both bottled waters on the market, changed the labels in the project office, and prepared the masked water for this study. The participants consumed 1 L of either water type, daily, for 12 weeks.

### 2.4. Evaluation

A self-administered questionnaire was used to assess the general gut health status of the participants. The following biomarkers were analyzed in the collected fecal samples: secretory immunoglobulin A (sIgA), five putrefactive products (phenol, *p*-cresol, 4-ethylphenol, indole, and skatole), and nine short-chain-fatty-acids (SCFAs, succinic, lactic, formic, acetic, propionic, isobutyric, butyric, 3-methylbutanoic, and valeric acids). The analyses were performed at TechnoSuruga Laboratory Co., Ltd. (Shizuoka, Japan) [33,34]. Three isoflavones (genistein, daidzein, and equol) were analyzed in urine. We analyzed the data in these biomarkers with non-parametric analysis to assess the differences in the change between pre- and post-interventions. Based on the differences in the change in fecal biomarker concentrations through the intervention, we performed multiple logistic regression analysis to evaluate the relationship between these waters and biomarkers.

#### Self-Administered Questionnaire

All participants answered questions on their general gut health and eating habits, such as constipation, abdominal discomfort, medicine intake, and consumption of unusual foods and drinks. Constipation was defined according to World Gastroenterology Organization [35].

### 2.5. Measurement of Samples

#### 2.5.1. Measurement of Fecal sIgA

The feces (0.1 g) were placed in a 2.0 mL-tube with zirconia beads and suspended in a mixture of 0.1 mM perchloric acid with 3% phenol. Samples were heated at 80 °C for 15 min, vortex-mixed at 5 m/s for 45 s using FastPrep 24 (MP Biomedicals, Santa Ana, CA, USA), and then centrifuged at 15,350× *g* for 10 min. The supernatant was passed through a 0.45 µm filter. The supernatant was employed for the measurements of sIgA and SCFAs. The sIgA levels were measured using Human IgA ELISA Quantitation kit (E80-102, Bethyl Laboratories Inc., Montgomery, TX, USA), according to the manufacturer’s instructions, and a microreader (Varioskan Flash, Thermo Fisher Scientific, Waltham, MA, USA).

#### 2.5.2. Measurement of Fecal Putrefactive Products

The feces (0.1 g) was suspended in 2.5 mL of phosphate buffer containing 0.4 mg/L 4-isopropylphenol as an internal standard. Samples were heated at 85 °C for 15 min, mixed with 2.5 mL of acetonitrile and 1 g NaCl, shaken for 30 min, and then centrifuged at 1300× *g* for 10 min. Then, 1 mL of the supernatant was dehydrated and purified by passing through a sodium sulfate drying cartridge (Bond Elut LRC, Agilent Technologies, Tokyo, Japan), a C18 cartridge (Smart SPE C18-30, AiSTI Science, Wakayama, Japan), and a PSA cartridge (Smart SPE PSA-30, AiSTI Science, Wakayama, Japan), and placed in a vial.

The indole and phenol levels were determined by gas chromatography/mass spectrometry (QP-2010, Shimadzu, Kyoto, Japan) and a capillary column (Inert cap WAX, 30 m × 0.25 mm × 0.25 µm, GL Science, Tokyo, Japan). Helium was used as the carrier gas at a flow rate of 1.11 mL/min. The injector temperature and the interface temperature were maintained at 240 and 230 °C, respectively. The oven temperature program was as follows: 70 °C for 2 min; then 20 °C/min to 200 °C, held for 3 min; 10 °C/min to 240 °C; and finally, 240 °C, held for 16 min. For the analysis, 1 µL of the extract was injected in the splitless mode. The mass spectrometer was operated in electron impact ionization mode at 70 eV. The measurements were recorded using an absolute calibration curve (range: from 0.02 to 5.0 mg/L). Data were acquired in the selected ion-monitoring mode for quantification.

#### 2.5.3. Measurement of Urinary Isoflavones

Genistein, daidzein, equol, and propyl 4-hydroxybenzoate were purchased from Cayman Chemical, Combi-Blocks, Wako Pure Chemical, and Daicel, respectively. β-glucuronidase/sulphatase type H-2 from *Helix pomatia* was purchased from Sigma-Aldrich Japan. All other chemicals were analytical grade or HPLC grade. Samples of urinary genistein, daidzein, and equol levels were prepared and measured as described elsewhere [36], with modifications. Briefly, urine (800 µL) was mixed with 80 µL of 1 M sodium acetate (pH 4.5) and 8 µL of β-glucuronidase/sulfatase solution (Sigma-Aldrich Japan), and incubated at 37 °C for 4 h to hydrolyze the conjugated forms of isoflavones. As an internal standard, 80 µL of propyl 4-hydroxybenzoate (100 µg/mL) was added the mixture, and the analytes were extracted with 1.5 mL of dichloromethane and evaporated at 50 °C. The residue was dissolved in 400 µL of methanol; 20 µL were injected into the HPLC system (Shimadzu Co., Ltd., Koto, Japan). The HPLC conditions were as follows: column, TSK-gel ODS-100V (4.6 mm i.d. × 250 mm, Tosoh Co., Ltd., Tokyo, Japan); mobile phase A, phosphate/water/ethyl acetate/methanol (0.4:800:18:182, *v*/*v*); mobile phase B, ethyl acetate/methanol (20:980, *v*/*v*); elution, a linear gradient of 0–60% mobile phase B in 30 min; flow rate, 1.0 mL/min; column temperature, 40 °C; UV detection, 280 nm. The detection limits were 100 ng/mL for genistein and daidzein, and 200 ng/mL for equol. The amounts of daidzein and genistein were corrected for the presence of creatinine, and are expressed as mg/g-Cre. The amount of equol was corrected for the presence of daidzein as equol/daidzein, and expressed as mg/mg daidzein.

### 2.6. Statistical Analysis

Participants were randomly assigned in two groups. We assessed the differences in pre-intervention biomarker concentrations between RDSW and mineral water groups allocated by non-parametric analysis with Mann–Whitney U test. We found no significant difference and these data were utilized in following analyses in this study (Table 1). For all biomarkers in this study, the measured values were statically compared between pre- and post-interventions in the two groups using the Wilcoxon signed-rank test (significance, *p* < 0.05), as appropriate. For fecal biomarkers, formic acid was excluded from the statistical analysis due to small size (*n* = 5). Based on the differences in the change in fecal biomarker concentrations through the intervention, we performed multiple logistic regression analysis to evaluate the relationship between these waters and fecal biomarkers (significance, *p* < 0.05). The multiple logistic regression analysis was performed using adequate data without the extreme values. The proportion of the responder with increased change in fecal biomarker concentrations through the intervention was analyzed using the χ^2^-test (significance, *p* < 0.05). For the urinary isoflavones, subjects with detectable (≥200 ng/mL) and undetectable equol levels were classified as equol producers and non-producers, respectively. Equol non-producers were excluded from the statistical analysis for equol level assessment. All analyses were performed with BellCurve for Excel ver. 3.20 (Social Survey Research Information Co., Ltd., Tokyo, Japan).

## 3. Results

### 3.1. Self-Administered Questionnaire

The questionnaire (*n* = 98) revealed that 17 and 15 subjects in the RDSW and control groups, respectively, suffered from constipation prior to the study per the definition of constipation. Drinking water ameliorated the symptoms of constipation in 16/17 (94%) and 9/15 (60%) individuals in the RDSW and control groups, respectively. The improvement rate was significantly higher in the RDSW group than that in the control group (Figure 2).

### 3.2. Analysis of Fecal sIgA, Putrefactive Products, and SCFAs

Three out of 49 subjects in the control group were excluded because of an insufficient volume of feces. Consequently, samples for 49 subjects from the RDSW group and 46 subjects from the control group (mean age: 44 years; range: 22–71 years; 19 men and 27 women) were included in the fecal analysis.

The average values of fecal biomarker concentrations through the intervention are summarized in Table 2. The concentrations of sIgA and phenol significantly decreased in post-intervention in the RDSW group per the Wilcoxon signed-rank test. However, we found no statistically significant differences in the change between pre- and post-intervention in the control group.

Regarding to SCFAs, the differences in the change in SCFAs concentrations through the intervention were analyzed using the Wilcoxon signed-rank test. These demonstrated that five SCFAs (succinic, lactic, acetic, isobutyric, and 3-methylbutanoic acids) significantly increased post-intervention in the RDSW group. An increased change in propionic acids was observed in the RDSW group without any statistically significant difference. By contrast, the concentrations of two biomarkers (acetic acid and propionic acid) significantly decreased post-intervention in the control group. During this study, the combined amount of nine SCFAs increased from 5.81 to 6.3 mg/g in the RDSW group and decreased from 5.92 to 5.05 mg/g in the control group (Table 2, Figure 3a). The combined amounts of SCFAs through the intervention significantly increased in the RDSW group compared with the control group, with a 23% difference between two groups (Figure 3b). The proportion of subjects with increased SCFAs concentrations post-intervention is shown in Figure 4a. For all SCFAs, except for formic acid, the proportion of the responders was higher in the RDSW group than in the control group. The proportion of acetic acid responder was significantly higher in the RDSW group than in the control group. Next, the proportion of the responders by sex showed the differences between men and women (Figure 4b). The proportion of acetic acid responder was higher in men than in women in the RDSW group. However, the proportion of isobutyric acid and 3-methylbutanoic acid responders was higher in women than in men in the RDSW group. By contrast, we found no significant difference in the control group. In addition, based on the differences in the change in fecal biomarker concentrations through the intervention, multiple logistic regression analysis was performed to evaluate the relationship between these waters and fecal biomarkers. These revealed that RDSW significantly affected only two biomarkers (acetic acid and 3-methylbutanoic acid). These observations indicated that drinking RDSW more noticeably impacted SCFA production in the human gut ecosystem than drinking mineral water. Formic acid was detected in samples from only 5 out of 95 subjects, in a range of 0.01–0.02 mg/mL (the detectable limit was 0.01 mg/mL), suggesting that only minute amounts of formic acid are produced in the human intestine.

### 3.3. Analysis of Urinary Isoflavones

Samples from one subject from each group were excluded because of an insufficient volume of urine (Figure 1). Consequently, samples from 48 subjects from the RDSW group (mean age: 48 years; range: 27–72 years; 18 men and 30 women) and 48 subjects from the control group (mean age: 44 years; range: 22–71 years; 21 men and 27 women) were included in the urinary isoflavone analysis. Daidzein was not detected in the urine from two subjects in the RDSW group, and genistein was not detected in the urine from one subject from each group. No daidzein or genistein were detected in the urine from one subject from the RDSW group. The overall detection rate of both urinary daidzein and genistein was 97% (93/96). Equol was detected in the urine from 19 and 17 subjects in the RDSW and control groups, respectively. In the current study, equol was identified in 38% of subjects (36/96).

The detection rates of urinary isoflavones are summarized in Table 3. If urinary isoflavones were not detected in a sample, the value was considered zero. The average changes in the concentrations of the three isoflavones through the intervention were compared between the two groups. The corrected daidzein and genistein concentrations increased in the RDSW group (1.44 and 1.05, respectively) but decreased in the control group (–0.14 and –0.39, respectively; Figure 5a). Considerable amounts of both isoflavones were detected in the RDSW group. Equol concentrations were evaluated in 16 subjects from each group, following an elimination of three and one subjects from the RDSW and control groups, respectively, due to the lack of detection of daidzein at either point throughout the intervention (Table 3). Overall, increased average equol concentrations were observed in both groups (0.42 and 0.18 in the RDSW and control groups, respectively; Figure 5b). The equol production in the RDSW group was at least two times higher than that in the control group; however, the difference was not statistically significant.

## 4. Discussion

In the current study, we evaluated the effect of prolonged intake of RDSW on the gut health in humans, as assessed using a questionnaire and the changes in fecal and urinary biomarker concentrations of microbial metabolites both pre- and post-intervention. Drinking RDSW was more beneficial than drinking conventional mineral water, significantly increased the concentrations of five SCFAs, and relieved constipation. These changes support the use of RDSW as a beverage that promotes intestinal and systemic health.

Moderate and balanced food and liquid intake is essential for a healthy life in all age groups, and can eliminate the risk factors or prevent the onset of disease. The quality of processed and natural foods, including ocean-related products, is gaining public attention. The ocean, occupying approximately 70% of the Earth surface, supports all life on Earth as a source of food and energy. We previously reported that food components and certain natural products from the Kochi Prefecture (Japan), such as Chinese chive [37] and RDSW produced from the DSW in the Muroto promontory, provide important health benefits [38]. In particular, RDSW, a mineral-rich drinking water, exerts potentially beneficial effects on human health, as demonstrated by previous clinical studies [1,2,3,4,5,6,7,8,9,10,11,12,13,14,15,16,17].

In the current study, we showed that drinking RDSW effectively increases the amount of fecal SCFAs post-intervention. The concentrations of five SCFAs (succinic, lactic, acetic, isobutyric, and 3-methylbutanoic acids) significantly increased post-intervention in the RDSW group but not in the mineral water group. An increased concentration of propionic acid was observed without statistical significance. The proportion of the eight SCFAs in the RDSW group was higher than in control group. The proportion of acetic acid was significantly different between the two groups. A sex gap in the participants was observed in acetic acid, isobutyric acid, and 3-methylbutanoic acid in the RDSW group. SCFAs widely affect the interplay between the host and gut microbiota, impacting human health [39]. The effect of drinking RDSW depends on the gut microbial environment, indicating that RDSW contributes to the maintenance of the gut ecosystem. The multiple logistic regression analysis revealed that drinking RDSW significantly affected two SCFAs (acetic acid and 3-methylbutanoic acids). Acetate is the predominant fecal SCFA, followed by propionate and butyrate [40,41]. This study showed that drinking RDSW but not mineral water increased the fecal amounts of acetate and propionate. Acetate promotes the defense function of the host epithelial cells against pathogenic infection, i.e., inhibition of the translocation of toxins [42]. Propionate stimulates the release of the anorectic gut hormone peptide YY and glucagon-like peptide-1 by human colonic cells. These hormones reduce appetite and prevent long-term weight gain in humans, helping lead to a healthy life [43]. SCFAs generated by fermentation of dietary fibers by the gut microbiota, particularly acetate and propionate, have been implicated in a variety of beneficial effects on the host, including trophic and anti-inflammatory effects on the gut [44]. This study demonstrated that drinking RDSW increased the concentrations of six among nine SCFAs, indicating that RDSW seems to induce the balanced production of SCFAs. SCFAs positively impact human health and prevent a variety of disorders acting via chemoattractant receptors expressed in diverse organs and cells, even in the neurons in the brain [45]. It is generally accepted that SCFAs act as an energy source and stimulate the colon epithelium, which facilitates the peristaltic motion. In the current study, constipation was alleviated to a significantly greater extent in the RDSW group that was accompanied by increased SCFA production than in the mineral water group. Alleviation of constipation probably led to decreased concentrations of phenol and sIgA in the RDSW group. At the very least, the increased SCFA levels might be involved in the diverse beneficial effects of RDSW mentioned above. However, the amounts of fecal SCFAs do not always reflect their amounts absorbed by the body. Direct comparison of the amounts of SCFAs absorbed by the body in the two groups would clarify the effectiveness of the RDSW treatment. This study was restricted by duration and budget. Thus, a large-scale clinical study would provide more details about the effects of RDSW.

Most Fabaceae contain copious amounts of isoflavones, such as genistein and daidzein. Aglycone-type isoflavones produced by the gut microbiota exert physiological effects on various aspects of health, including cancer, osteoporosis, cardiovascular diseases, and climacteric symptoms [46,47,48,49,50,51,52,53,54,55,56,57]. Equol is produced from daidzein by specific members of the gut microbiota. In western populations, the equol producers are estimated to be 20–30% of the total compared with 40–60% in Asian countries depending on the existence of specific gut bacteria and the intake of soy foods [25,58]. In the current study, the proportion of equol producers was 38% (36/96), similar to data from previous Japanese studies [58,59,60]. After RDSW intake, the urinary levels of equol increased, with the change being at least two times higher than in the mineral water intake group. The increases in the two other isoflavones (daidzein and genistein) were observed in only RDSW group and showed the same tendency as equol, without statistical significance.

In general, consumption of soy foods rich in daidzein does not lead to a conversion of an equol non-producer to an equol producer [61,62]. However, an equol non-producer can convert to an equol producer after consuming soy foods for more than four months [63,64], indicating the existence of unidentified bacteria that convert daidzein to equol even in the gut of equol non-producers. Iino et al. [58] identified novel bacteria that contribute to equol production and demonstrated that some equol non-producers that harbor such bacteria are converted to equol producers after a short period of isoflavone consumption. These studies highlighted the possibility that compounds such as isoflavones can stimulate the abundance and/or activity of unidentified bacteria that participate in equol production. In the current study, we did not observe conversions to equol production even after 12 weeks of RDSW ingestion. Further investigations are needed to identify new bacteria that contribute to equol production using next-generation sequencing of feces. However, drinking RDSW obviously increased urinary equol levels, indicating that it enhanced the activity of gut bacteria that participate in equol production. The creation of aglycone-type isoflavone, which is substantially absorbed by the body and functions physiologically [65], requires β-glucosidase activity. We cannot exclude the possibility that RDSW influenced the activity of β-glucosidase in the gut microbiota and/or gut environment to produce aglycone-type isoflavone. The differences in the mineral concentrations between RDSW (hard water) and mineral water (soft water) reflect the differences observed in this study. Thus, a clinical study with medium-harness water (approximately 100 hardness) would provide more insight into the human gut ecosystem with respect to mineral concentrations.

Throughout the study, we strictly monitored the dietary habits of the participants to ascertain common lifestyle, without consumption of unusual foods, supplements, or medicines. Taken together, the findings of the current study indicated that RDSW intake fundamentally promotes the induction of these metabolites on the gut environment. The habit of eating soy foods and RDSW is a promising dietary approach to effectively maintaining human health by supporting the gut ecosystem.

## 5. Conclusions

The presented clinical study revealed that RDSW intake effectively relieves constipation and increases fecal SCFA levels and urinary isoflavone levels, including equol levels. These observations indicated that drinking RDSW potentially improves the gut ecosystem and these beneficial effects extend to human health.

## Figures and Tables

**Figure 1 nutrients-12-02646-f001:**
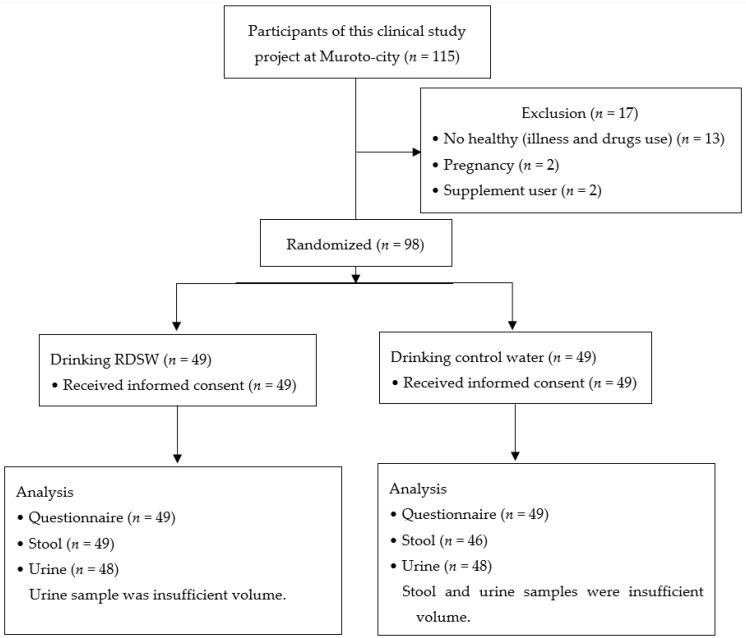
A total 98 healthy adults were enrolled from 115 adults who participated in this clinical study (industry–academia–government collaboration project) held in Muroto-city, Kochi, Japan. The subjects were healthy adults without any illnesses, and no prescription and commercial drugs use. Pregnant persons and supplement users were excluded from this study. RDSW, refined deep-sea water.

**Figure 2 nutrients-12-02646-f002:**
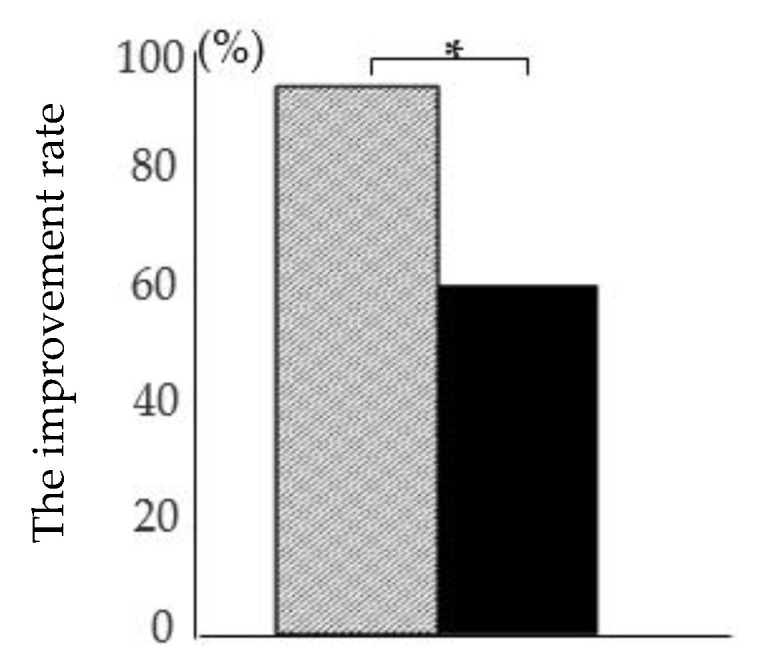
Alleviation of constipation by drinking RDSW or mineral water. In the study, 17 subjects (RDSW group) and 15 subjects (control group) reported constipation in the questionnaire. The constipation improvement rate was 94% and 60% in the RDSW group and control group, respectively. Hatched bar, RDSW group; black bar, control group; * *p* < 0.05.

**Figure 3 nutrients-12-02646-f003:**
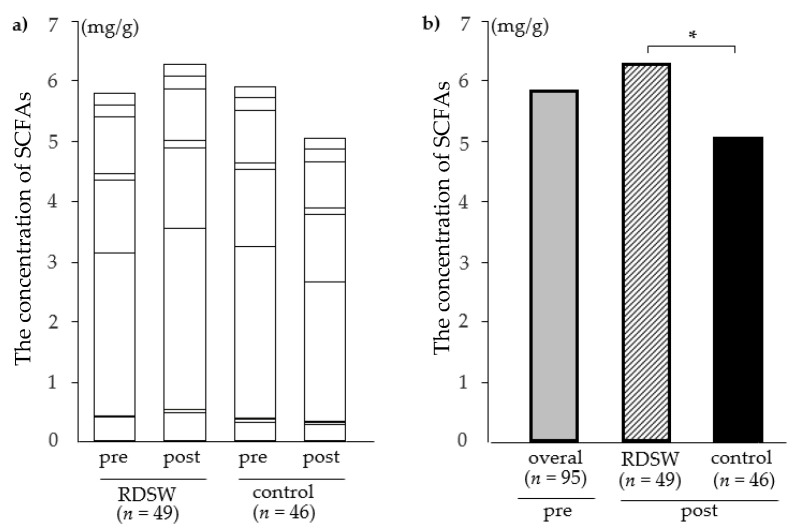
Effect of drinking RDSW or mineral water on fecal short-chain-fatty-acids (SCFAs) concentrations through the intervention (average). (**a**) The combined amounts of nine SCFAs increased in the RDSW group and decreased in the control group. The following nine SCFAs were analyzed, from bottom to top: succinic acid, lactic acid, formic acid, acetic acid, propionic acid, isobutyric acid, butyric acid, 3-methylbutanoic acid, and valeric acid. Formic acid is not listed due to its extremely small amounts. (**b**) The change in the combined amounts of SCFAs was observed with a 23% difference between two groups. Shaded bar, all subjects; hatched bar, RDSW group; black bar, control group; pre, pre-intervention; post, post-intervention; * *p* < 0.05.

**Figure 4 nutrients-12-02646-f004:**
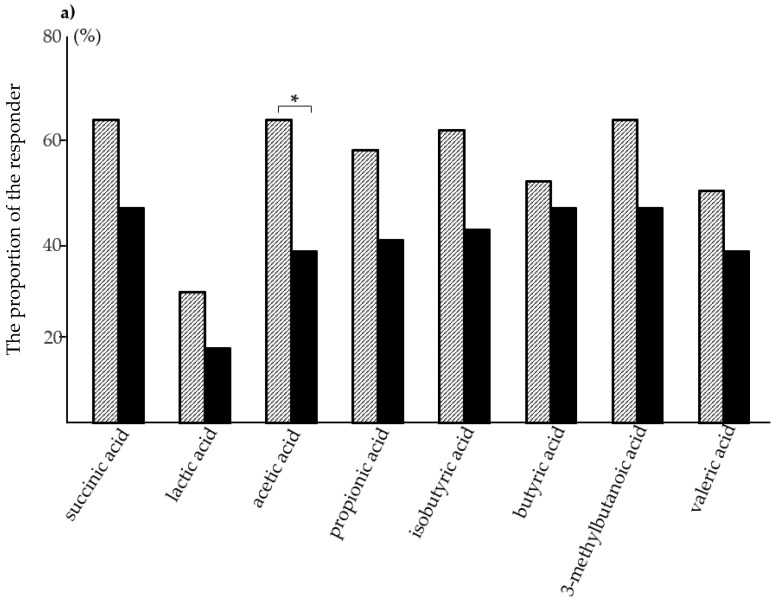
Participants with increased change in eight SCFAs concentrations post-intervention during the study period. Small amounts of formic acid were only detected in five subjects; ad therefore, formic acid was excluded from the analysis. (**a**) The proportion of eight SCFAs was higher in the RDSW group than in the control group. In particular, a significant difference in only acetic acid between the two groups was found. (**b**) The proportion of the participants by sex. Significant differences between men and women were observed in the RDSW group as follows: acetic acid responders in men, and isobutyric acid and 3-methylbutanoic acid responders in women. Hatched bar, RDSW group; black bar, control group; M, men; W, women; * *p* < 0.05.

**Figure 5 nutrients-12-02646-f005:**
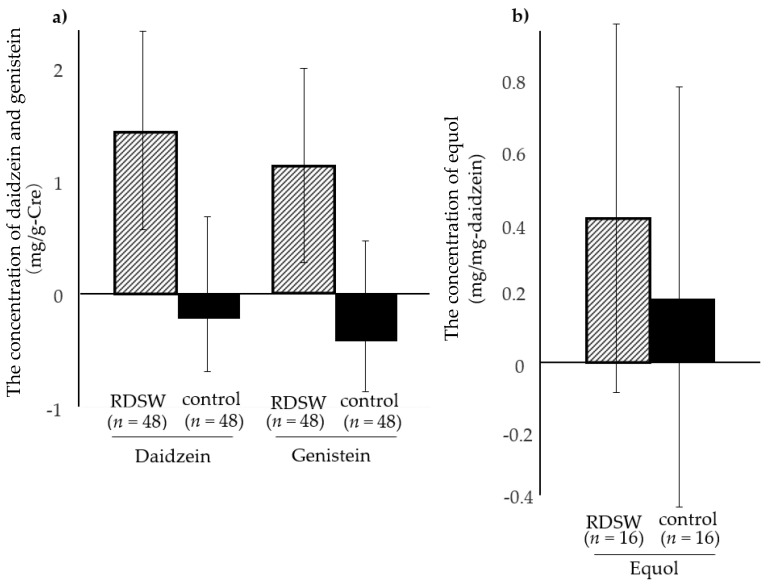
The differences in the change in the urinary biomarker concentrations through the intervention (average). Daidzein and genistein (**a**) and equol concentrations (**b**). (**a**) Data for 48 subjects from each group were analyzed. Daidzein and genistein concentrations were corrected for creatinine content, and are expressed as mg/g-Cre. (**b**) Data for 16 subjects from each group were analyzed. Equol concentrations were corrected for daidzein content, and are expressed as mg/mg daidzein. Hatched bar, RDSW group; black bar, control group; bar depicts SE (standard error).

**Table 1 nutrients-12-02646-t001:** Characteristics of participants in pre-intervention of two groups (average).

	RDSW			Mineral Water (Control)		
	Total (*n* = 49)	Men (*n* = 19)	Women (*n* = 30)	Total (*n* = 49)	Men (*n* = 21)	Women (*n* = 28)
Age (year)	48.1	45.9	49.4	44.2	44.2	44.2
BMI (kg/m^2^) (Body Mass Index)	24.8	26.8	23.6	24.1	25.6	23.1
	Total (*n* = 49)	Men (*n* = 19)	Women (*n* = 30)	Total (*n* = 46)	Men (*n* = 19)	Women (*n* = 27)
sIgA (μg/g) (secretory immunoglobulin A)	656	810	558	536	571	512
Putrefaction (μg/g)						
Phenol	6.01	10.02	3.47	5.72	9.63	3.68
*p*-Cresol	51.59	43.78	56.53	62.78	27.26	87.78
4-Ethylphenol	0.60	0.36	0.75	1.03	0.70	1.26
Indol	25.93	26.31	25.69	27.08	20.57	31.67
Skatol	2.69	2.31	2.94	3.69	2.72	4.37
SCFA (short-chain-fatty-acids, mg/g)						
Succinic acid	0.39	0.65	0.22	0.30	0.33	0.29
Lactic acid	0.02	0.04	0.01	0.06	0.12	0.02
Formic acid	0.00	0.00	0.01	0.00	0.01	0.00
Acetic acid	2.73	3.09	2.50	2.88	3.43	2.49
Propionic acid	1.22	1.48	1.05	1.29	1.40	1.21
Isobutyric acid	0.10	0.10	0.10	0.11	0.08	0.13
Butyric acid	0.95	0.98	0.93	0.87	0.86	0.87
3-Methylbutanoic acid	0.19	0.18	0.20	0.22	0.14	0.28
Valeric acid	0.20	0.21	0.19	0.18	0.16	0.20
	Total (*n* = 48	Men (*n* = 18)	Women (*n* = 30)	Total (*n* = 48)	Men (*n* = 21)	Women (*n* = 27)
Urine Isoflavones						
Daidzein (mg/g-Cre)	1.74	1.98	1.56	3.28	2.50	3.89
Genistein (mg/g-Cre)	2.16	2.34	2.04	3.99	3.27	4.55
	Total (*n* = 19)	Men (*n* = 6)	Women (*n* = 13)	Total (*n* = 17)	Men (*n* = 9)	Women (*n* = 8)
Equol (mg/g-Cre)	2.57	2.53	2.58	2.30	1.98	2.66

**Table 2 nutrients-12-02646-t002:** The values of fecal biomarkers of two groups through the intervention (average).

	RDSW (*n* = 49)		Mineral Water (Control) (*n* = 46)	
	Pre-Intervention	Post-Intervention	Pre-Intervention	Post-Intervention
sIgA (μg/g)	656	567 ↓ **	536	544
Putrefaction (μg/g)				
Phenol	6.01	3.08 ↓ *	5.72	5.77
*p*-Cresol	51.59	54.22	62.78	61.56
4-Ethylphenol	0.60	1.21	1.03	0.46
Indol	25.93	25.99	27.08	30.50
Skatol	2.69	2.65	3.69	3.69
SCFA (mg/g)				
Succinic acid	0.39	0.47 ↑ *	0.30	0.27
Lactic acid	0.02	0.05 ↑ *	0.06	0.03
Formic acid #	0.00	0.00	0.00	0.02
Acetic acid	2.73	3.04 ↑ *	2.88	2.34 ↓ **
Propionic acid	1.22	1.33	1.29	1.12 ↓ *
Isobutyric acid	0.10	0.13 ↑ *	0.11	0.12
Butyric acid	0.95	0.86	0.87	0.78
3-Methylbutanoic acid	0.19	0.23 ↑ *	0.22	0.21
Valeric acid	0.20	0.19	0.18	0.17

* *p* < 0.05; ** *p* < 0.01; ↑, increase; ↓, decrease; # formic acid was excluded from analysis.

**Table 3 nutrients-12-02646-t003:** Detection rates of urinary isoflavones in the two groups.

	RDSW (Rate, % Total)	Mineral Water (Rate, % Total)	Total (Rate, % Total)
Daidzein	46 */48 (96)	48 **/48 (100)	94/96 (98)
Genistein	47 ^#^/48 (98)	47/48 (98)	94/96 (98)
Daidzein and Genistein	46/48 (96)	47/48 (98)	93/96 (97)
Equol	19 ^†^/48 (40)	17 ^††^/48 (35)	36/96 (38)

* Including three subjects in whose samples daidzein was detected at either point throughout the intervention (pre- or post-intervention) of the study. ** Including one subject in whose samples daidzein was detected at either point through the intervention of the study. ^#^ No genistein or daidzein were detected in one subject. ^†^ Including three subjects in whose samples equol was detected at either point through the intervention of the study. ^††^ Including one subject in whose samples equol was detected at either point through the intervention of the study.

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
