# Peer review of "Drinking Refined Deep-Sea Water Improves the Gut Ecosystem with Beneficial Effects on Intestinal Health in Humans: A Randomized Double-Blind Controlled Trial"

_nutrients, 2020, doi:10.3390/nu12092646_

Round 1

Reviewer 1 Report

I have read the paper titled, "Drinking refined deep-sea water improves the gut ecosystem with beneficial effects on the intestinal health in human: a randomised double-blind controlled trial" with great enthusiasm. This is a very interesting research within my expertise. I have the subject matter and statistical expertise to evaluate the research.

I believe the authors have done a great job in terms of measuring different fecal and urine biomarkers pertinent to gut ecosystem. The randomised design is plus. Nevertheless, I feel the study design, statistical analyses and reporting sections did not follow the standard procedures of a randomized trial. I also believe the statistical analyses of their trial result is not accurate. My specific comments are below:

1) Authors have listed several outcomes, including the fecal and urine biomarkers and questionnaire information. Nevertheless, it is important to note what is the primary outcome, and what are the secondary outcomes. 

2) What is the basis of 98 sample size? For an RCT, authors need to report the assumption made for calculating the sample size based on the primary outcome?

3) It is very important authors report the CONSORT guideline for reporting an RCT result. http://www.consort-statement.org/

4) The authors mentioned about the double-blinded RCT without mentioning any procedures of masking or blinding.

5) It is not clear what type of analyses were done: intention-to-treat. per-protocol, or as-treated. The authors have dropped some participants in analyses phase, which risk the breakdown of randomization.

6)My understanding is that the pre-treatment biomarkers were used to demonstrate the result of the intervention. Before-after analyses of an intervention is not the appropriate analyses for an RCT. The pre-treatment findings should be used to demonstrate whether the randomization worked or not. Some of the biomarkers listed in Table 1 suggest that substantial differences in pre-intervention biomarker concentration for RDSW and Mineral water arms. Does is mean that randomization did not work? A statistical test will be helpful. 

7) Comparisons need to be done between intervention and controlled arms only. Rationale of authors to consider a parallel group trial instead of a cross-over trial? 

8) Should the authors adjust the result for confounders such as age, sex, BMI and other participant-level covariates?

Author Response

1) Authors have listed several outcomes, including the fecal and urine biomarkers and questionnaire information. Nevertheless, it is important to note what is the primary outcome, and what are the secondary outcomes.

>Thank you for your comment. We aim to evaluate the effect of RDSW on the gut ecosystem in clinical trials with limitations. We focus the changes of faecal biomarkers (through the experiment) as the primary outcome, and questionnaire information (the improvement of constipation) and urine biomarkers are considered as the secondary outcomes. Based on these outcomes, we would like to provide new insights to understand the broad beneficial effects of drinking RDSW. We described these contents in appropriate positions (abstract, clinical study design…).

2) What is the basis of 98 sample size? For an RCT, authors need to report the assumption made for calculating the sample size based on the primary outcome?

>I understand what you pointed out the sample size. I agree that the more participant the better. If we could collect more samples, we must show more evidences of the effect of RDSW. However, we had to perform this clinical trial with severe restrictions of budget, fixed period of time, and others. Thus, approximately 100 participants were maximum size in this study as far as we could. We explained the concept and situation of this study in the ethics committees of university, and they understood and permitted this study as feasible study. I appreciate if you understand the situation. We briefly described these contents in appropriate positions (clinical study design, discussion…).

3) It is very important authors report the CONSORT guideline for reporting an RCT result. http://www.consort-statement.org/

>Thank you for your comment. According to your advice, we made new figure as the flow-diagram (Figure 1). I hope that it is helpful for you and all readers in this journal.

4) The authors mentioned about the double-blinded RCT without mentioning any procedures of masking or blinding.

>We purchased both bottled waters in the market, changed the labels in the project office, and prepared the masked water for this study. Thus, nobody knows the source of the bottled water consumed during the experiment. We described these contents with the information of water in appropriate positions (ingestion schedule).

5) It is not clear what type of analyses were done: intention-to-treat. per-protocol, or as-treated. The authors have dropped some participants in analyses phase, which risk the breakdown of randomization.

>I apologized for your confusion. We focused the differences in the changes in biomarker concentrations between pre- and post-intervention. According to your advice, we re-analysed all adequate data by non-parametric statistical analyses such as Mann-Whitney and Wilcoxon, whenever appropriate. We described these contents with new table in appropriate positions (statistical analysis, results, discussion…).

>I agree with your comment about “decreased number becomes the risks for assessment of the study. “ I understand that the more participant the better (to avoid such risk). As mentioned above, however, we had to perform this clinical trial with severe restrictions. Thus, approximately 100 participants were maximum size in this study as far as we could. I appreciate for your understanding.

6) My understanding is that the pre-treatment biomarkers were used to demonstrate the result of the intervention. Before-after analyses of an intervention is not the appropriate analyses for an RCT. The pre-treatment findings should be used to demonstrate whether the randomization worked or not. Some of the biomarkers listed in Table 1 suggest that substantial differences in pre-intervention biomarker concentration for RDSW and Mineral water arms. Does is mean that randomization did not work? A statistical test will be helpful.

>Thank you for your comments and good suggestions. According to your suggestions, the differences in pre-intervention biomarker concentrations for RDSW and Mineral water arms were assessed by non-parametric analysis with Mann-Whitney test. These showed no statistical difference between the two groups allocated and these data were utilized in following analyses in this study. Furthermore, we re-analysed all data in faecal biomarkers as primary outcome with non-parametric analyses to assess the differences in the change between pre- and post-intervention in this study. In addition, based on the differences in the change in faecal biomarker concentrations through the intervention, we preformed the logistic regression analysis to evaluate the relationship between these waters and faecal biomarkers. The multiple logistic regression analysis without the extreme values was performed. Thus, we described these contents with new table 2 in the appropriate positions (subjects, statistical analysis, ,results, discussion….). Thank you again for your good suggestions.

7) Comparisons need to be done between intervention and controlled arms only. Rationale of authors to consider a parallel group trial instead of a cross-over trial?

>According to your suggestion, I re-analysed the differences in any items including biomarker concentrations between pre- and post-intervention by non-parametric analyses as mentioned above. I understand what you pointed out the study design. At first, we planned the cross-over design. However, in the fact, it was difficult for participants due to log-term experiment (at least 10 months). We predicted many participants would drop out during the experiment of cross-over study. In addition, the study was severely restricted with budget and seasonable condition based on their lifestyles. I appreciate for your understanding the restricted situations of this study.

8) Should the authors adjust the result for confounders such as age, sex, BMI and other participant-level covariates?

>According to your suggestion. We performed the multiple logistic regression analysis as mentioned above, demonstrating that RDSW significantly affected only 2 biomarkers (acetic acid and 3-methylbutanoic acid) with p < 0.05. These indicated that drinking RDSW positively affected to increase the concentration of these biomarkers. No more effective factor related with these waters was found by the analysis we could. We described these contents in appropriate positions (statistical analysis, results, discussion…). I appreciate for your understanding.

Reviewer 2 Report

The study of Takeuchi et al. determined the effect of drinking refined deep-sea water (RDSW) on the gut ecosystem compared with mineral water in a randomised double-blind controlled trial. After 12 weeks of drinking, constipation was significantly alleviated and total short-chain fatty acid levels in feaces were increased in the RDSW group. In addition, isoflavone levels were increased in the urine of RDSW group. They conclude that drinking RDSW improves intestinal environment in humans.

Major comments:

  • In some cases, words/ sentences are not appropriate, e.g. ´worry-free` (line 43), ´available on the market` (line 44), ´hot research topics`(line 79) etc.
  • Participants are not characterized well. Characteristics, e.g. age, gender, body weight, some clinical parameters etc. of the particapants summarized in a table will be helpful. Which inclusion and exclusion criteria were defined?
  • The authors mentioned that consumption of food was assessed. What about other beverages (e.g. coffee, tea, alcohol, sugar-sweetened beverages etc) besides 1 L of RDSW or mineral water? It might be of interest to assess caloric intake, macronutrients and mineral and vitamins of the probands.
  • What was the composition of mineral content of RDSW and mineral water?
  • The authors found out that constipation was improved. Was this a main outcome of the study?
  • It will be interesting if and how microbiota composition of the participants changed? Was microbiota composition analysed?
  • Labelling of axis of figures is not complete.
  • Statistics: Graphs did not show SEM or SD.
  • Discussion can be in more depth.

Author Response

In some cases, words/ sentences are not appropriate, e.g. ´worry-free` (line 43), ´available on the market` (line 44), ´hot research topics`(line 79) etc.

>Thank you for your comment. Basically, the manuscript was checked by native (company) with certification. However, we revised the manuscript according your suggestions as far as we could. Furthermore, we added more results obtained from statistical analyses and described these in appropriate positions of the manuscript. I appreciate for your understanding.

Participants are not characterized well. Characteristics, e.g. age, gender, body weight, some clinical parameters etc. of the particapants summarized in a table will be helpful. Which inclusion and exclusion criteria were defined?

>Thank you for your good comment. According to your suggestions, I summarized the characteristics of participants and made new table 1. Participants are healthy adults without any illness and any drugs use. In addition, pregnancy and supplement user were excluded. I made the flow-diagram of this study (Figure 1). I hope these table and figure would be helpful.  

The authors mentioned that consumption of food was assessed. What about other beverages (e.g. coffee, tea, alcohol, sugar-sweetened beverages etc) besides 1 L of RDSW or mineral water? It might be of interest to assess caloric intake, macronutrients and mineral and vitamins of the probands.

>Thank you for your good comment. I understand and agree with what you mean about other beverages. At first, we planned to include these beverages in this study as factors. However, it was difficult in this study (sample size) with restricted budget, term of study and others. Thus, this clinical trial was performed based on their normal/common lifestyles; however, we strictly monitored the keeping of their normal/common lifestyles during the study. Participants did not intentionally intake/consume any foods, beverages, supplements, and others beyond normal daily lifestyles. I appreciate for your understanding the restricted situations of this study.

What was the composition of mineral content of RDSW and mineral water?

>Thank you for your comment. I briefly described the information of RDSW (Amami no Mizu, Hardness 1000) in materials and methods (ingestion schedule) with reference [15]. I appreciate if you refer the reference. The mineral water is the most popular water consumed in Japan. I could not show the name and composition of the mineral water used with social consideration for the company. However, based on concentrations of magnesium and calcium, the hardness of the mineral water was calculated as 21. RDSW is hard water (hardness 1000) and mineral water is soft water (hardness 21). We added the information as far as we could (ingestion schedule). In addition, we briefly described the differences in nutrition concentrations between two waters in discussion. I appreciate for your understanding the situations.

The authors found out that constipation was improved. Was this a main outcome of the study?

>I apologized for your confusion about outcome. We focus the changes of faecal biomarkers (through the experiment) as the primary outcome, and questionnaire information (the improvement of constipation) and urine biomarkers are considered as the secondary outcomes. Based on these outcomes, we would like to provide new insights to understand the broad beneficial effects of drinking RDSW. We described these contents in appropriate positions (abstract, clinical study design…). I appreciate for your understanding.

It will be interesting if and how microbiota composition of the participants changed? Was microbiota composition analysed?

>Thank you for your good comment. I am remarkably interesting to what you mean about microbiota composition, too. In the fact, we obtained the results from research institute (company). These shows the differences in the change in microbiota between pre- and post-intervention in two types of water. However, we cannot complete to analyze the huge data yet. We would like to summarize and submit in next opportunity. I appreciate for your understanding.    

Labelling of axis of figures is not complete.

>According to your comments, I corrected the figures.

Statistics: Graphs did not show SEM or SD.

>According to your comment, I added bar as SE in figures.

Discussion can be in more depth.

>We re-analysed all data in faecal biomarkers with non-parametric analyses to assess the differences in the change between pre- and post-intervention in this study. In addition, based on the differences in the change in faecal biomarker concentrations through the intervention, we preformed the multiple logistic regression analysis to evaluate the relationship between these waters and faecal biomarkers. The multiple logistic regression analysis was performed using adequate data without the extreme values. We described these contents with new table 2 in the appropriate positions (abstract, methods, results, discussion). I appreciate for your consideration.    

Round 2

Reviewer 2 Report

All my comments were addressed. 

Author Response

According to your suggestions, the manuscript was again edited by the company (Language editing) MDPI recommends through on web site. In addition, I revised the statement in the final paragraph of Introduction as follows; The aim of the current clinical trial was to investigate whether RDSW influences the biomarkers associated with the human gut ecosystem, to provide new insights into the understanding of the broad beneficial effects of drinking RDSW.

I appreciate for your understanding.